# Housefly Pupae-Derived Antioxidant Peptides Exerting Neuroprotective Effects on Hydrogen Peroxide-Induced Oxidative Damage in PC12 Cells

**DOI:** 10.3390/molecules24244486

**Published:** 2019-12-07

**Authors:** Tingting Sun, Sichen Zhang, Wenzhe Yang, Zhimin Zhao, Depo Yang

**Affiliations:** 1School of Pharmaceutical Sciences, Sun Yat-sen University, Guangzhou 510006, China; suntt8@mail2.sysu.edu.cn (T.S.); zhangsch3@mail2.sysu.edu.cn (S.Z.); zhaozhm2@mail.sysu.edu.cn (Z.Z.); 2Guangdong Technology Research Center for Advanced Chinese Medicine, Guangzhou 510006, China; ywz19890409@sina.com

**Keywords:** housefly pupae, alcalase, antioxidant peptide, PC12 cells, neuroprotection

## Abstract

In this study, two antioxidant peptides were identified and characterized from the alcalase-hydrolysate of housefly (*Musca domestica* L.) pupae guided by ABTS cation radical scavenging activity. Peptides sequences were identified as DFTPVCTTELGR (DR12, 1338.48 Da) and ARFEELCSDLFR (AR12, 1485.66 Da) using nano-liquid chromatography-tandem mass spectrometry (LC-MS/MS). Both DR12 and AR12 exert strong ABTS cation radical scavenging ability with EC_50_ values of 0.39 and 0.35 mM, respectively. Moreover, AR12 can effectively protect PC12 cells from oxidative damage induced by hydrogen peroxide (H_2_O_2_) by decreasing intracellular reactive oxygen species (ROS) and malonaldehyde (MDA), recovering cellular mitochondrial membrane potential (MMP), and increasing the activity of intracellular superoxide dismutase (SOD). Stability tests suggest that AR12 is competent for the challenge of heating, acid, alkali or simulated gastrointestinal (GI) digestion and exhibits great activity to remove ABTS cation radical. DR12 shows a great stability against heating, but its antioxidative ability declines after being treated with acid, alkali or simulated GI digestion. In general, both DR12 and AR12 identified from housefly pupae hydrolysate stand a chance of being potential antioxidants or precursors to antioxidants and AR12 might be applied in the field of neuroprotection.

## 1. Introduction

Reactive oxygen species (ROS) are produced during cellular oxygen metabolism and play vital roles in vivo, such as inducing cell differentiation, proliferation and migration, and participating in inter- and intra-cellular signal transmission by stimulating glucose to be transported into cells [1]. However, when the body is suffering from fatigue or illness, the in vivo redox equilibrium could become disrupted, and the excessive accumulation of reactive species would lead to cellular oxidative stress and cause damage to various cellular components, such as membrane structures, DNA or proteins [2,3]. This is believed to be a primary cause or a secondary complication of various chronic diseases, e.g., cancer [4], diabetes [5], cardiovascular disease [6,7], rheumatoid arthritis [8], neurodegeneration, and aging process [9,10].

In the last two decades, there has been a marked increase in searching for antioxidant peptides from food products, animals or plants resources [11] due to their higher safety and activity compared to synthetic antioxidants, such as butylated hydroxyanisole (BHA) and butylated hydroxytoluene (BHT) which are reported to have latent acute toxicity, developmental [12] and reproductive toxicity [13]. Various studies have verified that these peptides are usually encrypted in the parent proteins without activity until they are released by various means like fermentation, enzymatic hydrolysis, and curing. For example, Zhu et al. identified an antioxidant pentapeptide (Gly-Lys-Phe-Asn-Val) from Jinhua ham, a kind of traditional marinated meat, and this peptide exhibited great radical scavenging activity as well as Fe^2+^ chelating ability [14]. A survey conducted by Wang et al. reported that LQAEVEELRAALE from duck meat hydrolysate showed strong DPPH radical scavenging activity [15]. Three new antioxidant peptides (SVL, EAVQ, and RDY) purified from mulberry (*Morus atropurpurea* Roxb.) leaf neutrase-hydrolysates displayed great hemolysis inhibition ability and cellular antioxidant activity [16]. Many other researchers have identified various novel antioxidant peptides in edible marine fishes, shellfish, and their processing by-products [17,18]. It is obvious that natural antioxidant peptides have a promising future in the pharmaceutical industry, food applications, and cosmetics field due to their abundant raw materials, high efficiency, and high safety.

Housefly (*Musca domestica* L.) is one of the best-known economic insects with many features like strong reproductive capacity, high adaptability, saprophagous peculiarity and pathogenic resistance ability. Housefly larvae and pupae, namely “wuguchong” in traditional Chinese medicine, are rich in biomass with high-quality proteins, polyunsaturated fats, chitosan, vitamins, minerals and other nutrients [19]. It has been used for more than hundreds of years in clinical practice to treat gastrointestinal disease and heal wounds like malnutritional stagnation, damp-heat diarrhea, vomiting, decubital necrosis, ecthyma and lip boils [20,21]. Recently, there have been several studies focusing on the investigation of antioxidant activity of wuguchong and some progress has been made. For example, researchers isolated the polypeptides hydrolyzed by neutral protease from housefly larvae and the peptides showed great protective effect on hydrogen peroxide (H_2_O_2_)-induced oxidative damage in HepG2 cells [22]; He et al. verified that housefly larvae powder could prevent oxidative stress injury via regulation of UCP4 and CyclinD1 and modulation of JNK and P38 signaling in APP/PS1 mice [23]. However, most of these studies were conducted based on mixtures of peptides, and few specific housefly peptides exerting antioxidant activity have been reported in existing studies. Moreover, housefly pupa is also rich in biomass, protein accounting for more than 60% of dried pupa, and our previous study indicated that the peptides derived from housefly pupae show strong antioxidant activity [24], but the exploration of active components of housefly pupae is almost blank to date. Therefore, this study was aimed to explore the antioxidant peptides of housefly pupae.

Based on existing researches and the issues set forth above, this study was designed to purify and identify antioxidant peptides from alcalase-hydrolysate of housefly pupae. Firstly, guided by ABTS cation radical scavenging assay, housefly pupae hydrolysate (HPH) was purified by cold acetone, ultrafiltration membranes with molecular weight cut-off (MWCO) of 10 kDa and 3 kDa, size exclusion chromatography (SEC) and reversed-phased high-performance liquid chromatography (RP-HPLC) in sequence. The fraction with the best activity was identified using nano liquid chromatography-tandem mass spectrometry (LC-MS/MS) and then the peptides were synthesized by solid-phase reaction. Subsequently, the neuroprotective effect of the synthetic peptides on H_2_O_2_-induced oxidative damage in PC12 cells was explored. Finally, the stability of peptides under thermal, pH and simulated gastrointestinal (GI) digestion treatments was assessed (Figure 1). 

## 2. Results and Discussion

### 2.1. Purification of Antioxidative Peptides

Several studies have clarified the contribution of molecular size and structural characteristics to peptides bioactivity [11,25]. These studies showed that the number of amino acids of most antioxidant peptides liberated from parent proteins by enzymatic hydrolysis usually ranged from 2 to 20, and the potent antioxidant peptides were generally enriched in low molecular weight fractions (<5 kDa). Ultrafiltration is a kind of membrane filtration to separate proteins according to molecular weight or concentration gradients under a certain amount of pressure [26].

In this study, the cold acetone-precipitated hydrolytic peptides were separated into three fractions (HPH-1, HPH-2 and HPH-3) with UF membranes with 3 and 10 kDa MWCOs. DPPH radical, ABTS cation radical and hydroxyl radical scavenging activities were used to evaluate the antioxidant capacities of each fraction. The results display that HPH-3 possesses higher radical scavenging ability than other fractions. In detail, the DPPH radical scavenging rate of HPH, HPH-1, HPH-2 and HPH-3 are 58.7 ± 7.1%, 51.5 ± 3.5%, 33.0 ± 4.2% and 81.5 ± 4.2% per 0.2 mg peptide, respectively. The ABTS cation radical scavenging rate of the aforesaid four samples are 54.5 ± 2.1%, 46.5 ± 2.3%, 29.0 ± 1.4% and 71.5 ± 4.9% per 0.5 mg peptide, respectively. The hydroxyl radical scavenging rate of the four samples are 49.0 ± 1.7%, 45.5 ± 2.1%, 24.5 ± 0.7% and 61.6 ± 3.8% per 2 mg peptide, respectively (Figure 2A,B). Thus, according to the above results, antioxidant activities of the three fractions are HPH-3 > HPH-1 > HPH-2 in descending order, it is obvious that the smallest fragment of molecular weight exhibits the best radical scavenging activity. However, it is also observed that the antioxidant activity does not show in descending molecular weight order. This result is slightly different from previous reports that the lower molecular size fraction had the higher antioxidant activities [19,26,27,28], but similar to the result reported by Wong et al. [27]. It could be conjectured that the peptides with lower molecular weight (HPH-3, <3 kDa) could react with free radicals much easier and convert them to more stable products and then terminate the radical chain reactions [29], which might make it clear that HPH-3 shows the best antioxidant activity. As for peptides with much higher molecular weight (HPH-1, > 10 kDa), they might contain more antioxidant activity sites compared to HPH-2 (3–10 kDa) that some sites could exhibit activity even in the parent protein. Therefore, according to the aforementioned results, HPH-3 with the best radical scavenging ability was chosen for further purification.

SEC is an efficient separation system on the basis of molecular size and widely applied to purify proteins and peptides, remove salt or exchange buffer of macromolecules solution [29]. As shown in Figure 2C, HPH-3 was fractionated into four subfractions (F1–F4), these subfractions were pooled according to their peak values and lyophilized to evaluate antioxidant activities. The result reveals that F4 has the highest ABTS cation radical scavenging ability (89.2 ± 1.1% per 0.2 mg peptide), while the scavenging rate of the other three fractions were below 50% per 0.2 mg peptide (Figure 2D).

F4 was further purified by RP-HPLC with a C18 column according to the structural properties of peptides, which is widely known as the final purification step due to its advantages of high sensitivity, resolution and column efficiency [30]. In general, large polar or hydrophilic peptides were eluted earlier, while the non-polar or hydrophobic peptides were eluted later. As displayed in Figure 2E, four peptide fractions (P1–P4) were obtained based on the elution time and concentrated in a rotary evaporator to evaluate their antioxidant activities. Among the four fractions, P3 with the retention time of 11.4 min shows the highest ABTS cation radical scavenging ability (Figure 2F), and the peak area accounts for 45.3% of the total peak area in the chromatogram at the absorbance 214 nm. In consideration of the highest antioxidant activity and the great purity, P3 was chosen for determination of peptides and their amino sequence.

### 2.2. Identification of Antioxidant Peptides

Two peptides DFTPVCTTELGR (Asp-Phe-Thr-Pro-Val-Cys-Thr-Thr-Glu-Leu-Gly-Arg, DR12) and ARFEELCSDLFR (Ala-Arg-Phe-Glu-Glu-Leu-Cys-Ser-Asp-Leu-Phe-Arg, AR12) were identified from P3 (Figure 3A,B). Molecular weights of synthetic DR12 (1338.48 Da) and synthetic AR12 (1485.66 Da) were consistent with the theoretical masses of 1338.49 Da and 1485.66 Da, respectively. Some studies have reported that DR12 is a part of peroxiredoxin 6 (Prx6) structure (42–53), which is an enzyme ubiquitously existing in life [31,32]. And several studies reported that AR12 is a part of heat shock proteins derived from human or dairy cows, respectively [33,34]. Although both of the sequence of DR12 and AR12 has been reported, their activities have not been characterized.

Previous studies have reported that the antioxidant ability of peptides is closely associated with some structural characteristic of these peptides, such as their molecular weight, amino acid compositions and sequences [35,36]. These studies clarified that the peptides with appropriately low molecular weight and containing hydrophobic amino acids, aromatic amino acids or some special amino acids like Cys residue can exert stronger antioxidant activity. In accordance with these views, the molecular weights of two peptides we identified were both between 500 Da to 1800 Da [37]. Additionally, DR12 contains four hydrophobic amino acid residues (Phe, Pro, Val, Leu) and an antioxidant amino acid residue (Cys), among which, Phe is aromatic. Similar to the above results, AR12 contains five hydrophobic amino acid residues including Ala, Phe, and Leu, and also contains a Cys residue, Phe is an aromatic amino acid. Hydrophobicity of peptides plays a key role in exerting antioxidant ability in vivo. Depending on their hydrophobicity, these peptides could smoothly get through membrane lipid bilayers or directly interact with lipids to terminate lipid peroxidation [35]. Furthermore, Pro could enhance the hydroxyl radical scavenging ability of peptides due to a pyrrolidine ring in its structure as a proton donor and Cys contains a disulfide bond which makes it to be the most active antioxidant amino acid [35]. Thus, the structure profiles of DR12 and AR12 are also consistent with the previous statements that hydrophobic amino acids, aromatic amino acids, and some antioxidant amino acids could make a great contribution for the antioxidant ability of peptides.

### 2.3. Neuroprotective Effect of Peptides on H_2_O_2_-Induced Cell Injury in PC-12 Cells

PC12 cells were employed to evaluate the cytoprotective effect of DR12 and AR12 against oxidative stress-induced damage. First of all, PC12 cells were treated with various concentrations of peptides to evaluate their potential cytotoxic effect using MTT assay. The result displays that both DR12 and AR12 have no cytotoxic effect even when the concentration is up to 100 µM (Figure 4B). Subsequently, the cells were dealt with different concentrations of H_2_O_2_ for 2 h. As we can see in Figure 4A, as the concentration of H_2_O_2_ increased, cell vitality decreases linearly and drops to 47.9 ± 2.3% when the concentration of H_2_O_2_ increases to 900 µM. In order to ensure that the effects of peptides could be better measured, 800 µM H_2_O_2_ was selected to induce oxidative damage of cells that there were at least half of viable cells. Figure 4C,D suggest that pretreatment with DR12 and AR12 could both recover the reduction of cells viability in a concentration-dependent manner, and AR12 shows better cytoprotective effect than DR12.

To further confirm the protective effect of AR12 against H_2_O_2_-induced cells injury, the percentage of cell apoptosis was measured by flow cytometry. In brief, the percentage of apoptotic cells induced by 300 µM H_2_O_2_ for 2 h is significantly increased from 5.2 ± 0.8% to 34.4 ± 3.3%. 

Nevertheless, pretreatment with various concentrations of AR12 reduces cellular apoptosis to 21.9 ± 1.1%, 12.1 ± 1.6% and 8.2 ± 2.8% in a dose-dependent manner (Figure 5). The above results indicate that the anti-apoptosis effect of AR12 could protect the PC12 cells from the damage induced by H_2_O_2_. However, DR12 shows little cellular anti-apoptosis effect under the same conditions. 

### 2.4. AR12 Protected H_2_O_2_-Induced PC12 Cells Damage through Reducing Intracellular ROS 

As the products of intracellular oxygen metabolism, ROS participates in cells signaling and tissues injury that contributes to many diseases’ progression, such as neurodegenerative diseases, aging process, and cardiovascular disease et al. [38]. Therefore, the level of intracellular ROS was detected in this study. As shown in Figure 6, intracellular ROS in PC12 cells bursts after incubation with 300 µM H_2_O_2_ for 2 h (fluorescence intensity increased from 0.5 ± 0.1% to 91.9 ± 3.9%). However, the oxidant burden of PC12 cells rapidly decreases in a dose-dependent manner after the pretreatment with various concentrations of AR12 for 24 h. These results imply that H_2_O_2_-induced ROS accumulation in PC12 cells could be effectively antagonized by AR12.

### 2.5. AR12 Protected H_2_O_2_-Induced PC12 Cells Damage through Recovering the Loss of Mitochondrial Membrane Potential (MMP)

Mitochondria play vital roles in various cellular biological processes, such as cell growth, cell cycle, apoptosis, and ROS generation. The function of mitochondria of cells could be indicated by MMP or mitochondrial morphology [10]. In order to determine the effect of AR12 on mitochondria of PC12 cells induced by H_2_O_2_, the MMP of PC12 cells treated with or without H_2_O_2_ or AR12 was detected by laser scanning confocal microscope. As displayed in Figure 7, exposure of cells to H_2_O_2_ leads to a sharp transformation of green fluorescence from red fluorescence, which indicates the collapse of MMP. While in AR12 pretreated groups, the intensity of green fluorescence is reduced in a dose-dependent manner, which suggests that AR12 is able to help recover the loss of cellular MMP to exert the neuroprotective effect.

### 2.6. Effects of AR12 on Malonaldehyde (MDA) and Superoxide Dismutase (SOD) in H_2_O_2_-Induced PC12 Cells 

As a product of lipid peroxidation, MDA is often regarded as a biomarker of oxidative stress. As expected, the MDA level in H_2_O_2_-induced cells significantly increase compared to control group, while the pretreatment with AR12 helps diminish the increase of MDA levels (Figure 8A). 

There are many antioxidant defenses in organisms including enzymatic antioxidant defenses (SOD, glutathione peroxidase (GPx), catalase (CAT)) and non-enzymatic antioxidant defenses (ascorbic acid (vitamin C), a-tocopherol (vitamin E), glutathione (GSH)). These antioxidant defenses could maintain the redox balance in vivo under normal conditions [2]. In this study, the cellular SOD activity was tested using WST-8 assay. As shown in Figure 8B, SOD activities decrease in the H_2_O_2_-induced cells. However, treatment with AR12 attenuates the loss of cellular SOD activities in a dose-dependent manner.

### 2.7. Effects of Thermal, pH and Simulated GI Digestion on Peptides

Thermal stability of antioxidants is critical for their production, processing, and transportation process, namely, the property of inactivation at high temperature greatly prevents antioxidants from wider applications. As shown in Figure 9A, ABTS cation radical scavenging ability of DR12 incubated at 25, 60, 70, 80, 90 °C almost keeps unchanged with EC_50_ values ranging from 0.36 mM to 0.38 mM. In contrast, the influence of heating on AR12 depicted in Figure 9B reveals that AR12 is sensitive to high temperature. In detail, the ABTS cation radical scavenging ability of AR12 significantly increases as treated at 60, 70, and 80 °C, followed by a decline at 90 °C, the EC_50_ values are 0.24, 0.23, 0.27, and 0.45 mM (vs 0.34 mM, treated at 25 °C), respectively. Theoretically, proteins after heat treatment would become denatured, condense, and then expose the hydrophobic domain. Additionally, hydrophobicity of peptides was reported to correlate with antioxidant properties. Therefore, it can be postulated that appropriately heating process could expose hydrophobic domain of AR12 and then make it easier to react with radicals. However, the excessive thermal treatment obviously destroys the active structure of AR12 that its ABTS cation radical scavenging ability significantly decreases after incubated at 90 °C compared with the untreated peptide. The aforementioned results suggest that both DR12 and AR12 could meet most heat treatment requirements, but excessive thermal process is not suitable for AR12. The influence of pH on peptides monitored by ABTS cation radical scavenging activity is displayed in Figure 9C,D. The EC_50_ values of DR12 markedly increase as pH below 3 or above 9, which may be resulted from the changes of charge in DR12, particularly at *N*- or *C*-terminal. However, AR12 displays favorable pH stability that the EC_50_ values range between 0.27 and 0.34 mM, interestingly, the ABTS cation radical scavenging ability of AR12 is always increased either in acidic or alkaline conditions. Both results of thermal and pH stability of AR12 could reveal that it can serve as a precursor for peptide with higher antioxidant activity.

Pepsin-pancreatin hydrolysis treatment was used to evaluate the effect of GI digestion on peptides (shown in Figure 9E,F). With 1 h of pepsin hydrolysis and 2 h of pancreatin hydrolysis at 37 °C, protease solution (pepsin and pancreatin) shows little activity to scavenge ABTS radical, DR12 has a slight decrease in ABTS radical scavenging ability by contrast to the activity of untreated peptide, which may be the result of degeneration of structure. However, AR12 shows great stability against the simulated GI digestion that there is no significant change in ABTS cation radical scavenging ability over the concentration change of AR12 between the untreated and treated peptide with EC_50_ values of 0.35 and 0.39 mM, respectively.

## 3. Materials and Methods

### 3.1. Materials and Reagents

Housefly (*Musca domestica* L.) pupae were obtained from Guangdong Magtech Bio-Technology Co., Ltd. (Guangzhou, China) and identified by Prof. Pang Hong from State Key Laboratory for Biocontrol and Institute of Entomology, Sun Yat-sen University. Ultrafiltration centrifugal units (MWCO of 3 and 10 kDa) were purchased from Merck (Darmstadt, Germany). Superdex 30 prep grade resins were purchased from GE Healthcare (Uppsala, Sweden). Acetonitrile (ACN) and trifluoroacetic acid (TFA) of HPLC grade were supplied by Macklin Biochemical Technology Co., Ltd. (Shanghai, China). 1,1-Diphenyl-2-picrylhydrazyl (DPPH), hydrogen peroxide (H_2_O_2_, 30%, *v*/*v*, 3-(4, 5-dimethyl-2-thiazolyl)-2, 5-diphenyl-2-*H*-tetrazolium bromide (MTT) and dimethyl sulfoxide (DMSO) were bought from Sigma-Aldrich (St. Louis, MO, USA). The differentiated rat pheochromocytoma (PC12) cell line was purchased from the Cell Bank of Shanghai Institute of Biochemistry and Cell Biology (Chinese Academy of Sciences, Shanghai, China). Roswell Park Memorial Institute (RPMI) 1640 Medium, fetal bovine serum (FBS), antibiotics (penicillin and streptomycin) and 0.25% trypsin-EDTA (phenol red) were bought from Gibco Life Technologies (Grand Island, NY, USA). Annexin V-FITC and propidium iodide (PI) Apoptosis Detection Kit was purchased from Becton, Dickinson and Company (BD, San Diego, CA, USA). Reactive Oxygen Species (ROS) Assay Kit, Mitochondrial Membrane Potential Assay Kit with JC-1, Total Malonaldehyde (MDA) Assay Kit, and Total Superoxide Dismutase (SOD) Assay Kit with WST-8 were purchased from Beyotime Biotechnology Co., Ltd. (Shanghai, China).

### 3.2. Preparation of HPH

Housefly pupae were collected and washed with distilled water, and then oven-dried at 50 °C for 2 days. The dried pupae were crushed to 24-mesh powder, defatted with *n*-hexane in a Soxhlet apparatus at 50 °C for 6 h, and then the defatted pupae powder was suspended in distilled water (*w*/*v*, 1:20) by stirring at 4 °C for 30 min to extract proteins. Whereafter, the mixture was heated in boiling water (100 °C) for 10 min to denature proteins. After cooling to 60 °C, the mixture was adjusted to pH 9.5. The hydrolysis reaction was conducted with alcalase amount of 6000 U/g at 60 °C for 30 min, following by heating to 100 °C for 15 min to inactivate the alcalase. Finally, the mixture was centrifuged at 12,000× *g* for 15 min and the supernatant (HPH) was collected for further purification.

### 3.3. Peptides Separation by Membrane Ultrafiltration

4-fold volume of cold acetone was added to HPH at 4 °C over the night to precipitate peptides, and then the resultant peptides were collected by centrifugation at 12,000× *g* for 15 min. Finally, the crude peptides were lyophilized and kept at 4 °C for further use. The crude peptides were fractionated sequentially using ultrafiltration membranes with MWCO of 10 kDa and 3 kDa at 4700× *g* for 30 min. The obtained three fractions (>10 kDa (HPH-1), 3–10 kDa (HPH-2), <3 kDa (HPH-3)) were freeze-dried and stored at 4 °C. The fraction with the best radical scavenging activity was selected for further purification.

### 3.4. Purification by Size Exclusion Chromatography 

HPH-3 was separated using a SEC column (26 mm × 60 cm) packed with Superdex 30 prep grade (34 µm, GE Healthcare Life Sciences). The column was equilibrated and eluted with 0.01 M phosphate buffer/0.15 M NaCl (pH 7.2). 1 mL of HPH-3 (50 mg/mL in mobile phase) was loaded on the column and eluted at the flow-rate of 3 mL/min at the room temperature and the monitoring absorbance was 214 nm. All fractions were collected and lyophilized to measure the content of peptides with the o-phthaldialdehyde (OPA) method [39] and evaluate their ABTS cation radical scavenging activity.

### 3.5. Purification by RP-HPLC

The eluted fraction of SEC with the highest antioxidative activity was further purified by RP-HPLC on an Ultimate AQ-C18 column (10 × 250 mm, 5 µm, Welch, Shanghai, China). 200 µL of F3 was injected and then eluted with a binary gradient where solvent A was water containing 0.05% TFA and solvent B was acetonitrile containing 0.05% TFA, the flow rate was set to 3 mL/min and the elution profile was detected at 214 nm and 280 nm. Gradient elution was performed as followed: 0–15% solvent B, 15 min; 15–100% solvent B, 5 min; 100% solvent B, 10 min; and 100–0% solvent B, 10 min. Eluted fractions were tested for ABTS cation radical scavenging activity.

### 3.6. Identification of Antioxidant Peptides by LC-MS/MS and Peptide Synthesis

Purified peptides (P3) were identified on a Thermo EASY-nLC 1200 system coupled to a Thermo Q Exactive Plus Orbitrap LC-MS/MS system (Thermo Scientific, Waltham, Mass, USA). Briefly, the peptides desalted by MonoTip C18 column (C18 solid phase extraction disk, 3M Company, St. Paul, MN, USA) were redissolved in 0.1% formic acid and loaded on an analytical column (75 µm × 30 cm, ReproSil-Pur 120 C18-AQ, 2 µm, Ammerbuch, Germany) under the constant pressure of 40 MPa. Eluent A was 0.1% formic acid aqueous solution and eluent B was acetonitrile/water/formic acid (80:20:0.1, *v*/*v*/*v*) solution, and then the gradient elution was performed at a flow rate of 200 nL/min, the elution program was as followed: 20–80% eluent B, 30 min; 80–100% eluent B, 30 min; 100% eluent B, 30 min. Mass spectrometry was performed with the positive ions electrospray scan mode (NSI source) and the scan range *m*/*z* 350–1700. The resolution of full MS was 70,000, and the resolution of the secondary mass was 35,000. Furthermore, the top ten most abundant doubly or multiply charged precursor ions in each MS scan were selected for fragmentation (MS2) by stepped higher energy collision dissociation (stepped HCD) of 25% around a normalized collision energy (NCE)-value of 20, 25, and 30. Thermo Scientific™ XCalibur™ 2.2 software (Thermo Scientific, Waltham, Mass, USA) was used to analyze the mass spectrum and the Thermo Proteome Discoverer 2.2.0.388 software (Thermo Scientific, Waltham, Mass, USA was used for homology searches between the obtained sequences against the Musca Domestica proteome database from Uniprot. Peptides identified from HPH were synthesized via the solid-phase peptide procedure by Jier biochemical Co., Ltd. (Shanghai, China). The synthetic peptides were purified by RP-HPLC with a Kromasil C18 colume (4.6 × 250 mm, 5 µm), purity of more than 98%, and the molecular weights were determined by LC/MS (LC/MS-2010 EV, Shimadzu, Kyoto, Japan).

### 3.7. Radical Scavenging Activity Assay

The DPPH radical scavenging activity of purified fractions was evaluated according to the published method with minor modifications [19]. Briefly, the sample group and blank group comprised 50 µL of the sample solution or vehicle mixed with 50 µL of 0.4 mM DPPH; and the control group comprised 50 µL of sample solution and 50 µL of ethanol (the vehicle of DPPH radical). The reaction was performed in the dark for 30 min. Subsequently, the scavenging of DPPH radical was determined by measuring the absorbance at 517 nm using a multifunctional microplate reader (FLUOstar Omega-ACU, BGM, Offenburg, Germany). The DPPH radical scavenging ability of the fractions was calculated according to the following equation:DPPH radical scavenging activity (%) = [1 − (*A_s_* − *A_c_*)/*A_b_*] × 100%(1)
where *A_s_*, *A_c_* and *A_b_* represent the absorbance of the sample, control and blank groups, respectively. Ascorbic acid was used as a positive control.

The ABTS cation radical scavenging activity assay was conducted according to the manufacture instruction. 10 µL of sample solution was added to 200 µL of fresh ABTS working solution, and the reaction mixture was incubated in darkness for 10 min, the absorbance was observed at 734 nm. The ABTS cation radical scavenging activity was calculated by the following formula: ABTS cation radical scavenging activity (%) = (1 − *A*_1_/*A*_0_) × 100%(2)
where *A*_1_ and *A*_0_ were the absorbances of sample group and control group, respectively. The sample solution of control group was replaced with deionized water. Ascorbic acid was used as a positive control.

The hydroxyl radical scavenging activity was determined on the basis of Zhou et al. [40] with some slight modifications. In simple terms, the reaction mixture contained 8 µL of 18 mM salicylic acid, 52 µL of sample solution, and 8 µL of 18 mM FeSO_4_, and then 32 µL of 0.1% H_2_O_2_ was subsequently added to the mixture to start the reaction. The absorbance was measured at 510 nm, and the scavenging activity was calculated using the following formula:hydroxyl radical scavenging activity (%) = [1 − (*A_n_* − *A_x_*)/*A_m_*] × 100%(3)
where *A_n_* was the absorbance of the sample group, *A_m_* presented the absorbance of blank group that the sample solution was substituted by deionized water, and *A_x_* was the absorbance of control group that FeSO_4_ was replaced by deionized water. Ascorbic acid was used as a positive control.

### 3.8. Cell Culture and Viability Analysis

The PC12 cells (PC12 cells were purchased from the Cell Bank of Shanghai Institute of Biochemistry and Cell Biology (Chinese Academy of Sciences, Shanghai, China). were cultured in RPMI 1640 medium supplemented with 10% FBS and 1% penicillin and streptomycin in a humid atmosphere of 5% CO_2_ at 37 °C.

PC12 cells were seeded at 5 × 10^3^ cells/well in a 96-well plate for 24 h, and then the cells were pretreated with various concentrations (100, 50, 25 µM) of peptides for 24 h. The medium was discarded and H_2_O_2_ (200–900 µM) was added to stimulate cells damage. After 2 h, 10 µL of 5 mg/mL MTT was added to each well and the cells were incubated for 4 h at 37 °C in the dark. Subsequently, the supernatant was removed and 100 µL of DMSO was added to dissolve the dark blue formazan crystals formed in intact cells, and the absorbance at 495 nm was measured with a microplate reader.

### 3.9. Apoptosis Assay

Apoptosis was assessed using the Annexin V-FITC and propidium iodide (PI) Detection Kit (BD) and detected by flow cytometer (Guava easyCyte, Millipore, Darmstadt, Germany) following the instructions of the manufacturer. Briefly, PC12 cells were seeded at 1 × 10^5^ cells/well in a 12-well plate for 24 h and pretreated with various concentrations (100, 50, 25 µM) of peptides for another 24 h, following by treatment of 300 µM H_2_O_2_ for 2 h. After the stimulation, the cells were harvested and washed with 1 × phosphate buffer saline (PBS) and resuspended in 1 × binding buffer, and then stained with Annexin V-FITC and PI for 15 min before detection. The analysis was done with FlowJo V10 software (BD, San Diego, CA, USA).

### 3.10. Intracellular ROS Level Measurement

ROS production in PC12 cells were detected with DCFH-DA by a laser scanning confocal microscope (FV 3000, Olympus, Tokyo, Japan) and a flow cytometry. DCFH-DA could be hydrolyzed by intracellular esterases and turns to be 2′,7′-dichlorofluorescin (DCF) with high fluorescence upon the oxidation by ROS. After drug treatment, the cells were incubated with 10 µM DCFH-DA in FBS-free medium at 37 °C for 20 min, and then the medium was aspirated and cells were washed twice with FBS-free medium. Finally, the images of cellular DCF fluorescence intensities were analyzed with FV31S-SW software and the quantitative analysis of fluorescence intensities were analyzed with FlowJo V10 software.

### 3.11. Measurement of MMP

MMP of PC12 cells was determined using JC-1 which could aggregate in mitochondrial matrix and formed J-aggregates with red fluorescence when the MMP was at a high state. In contrast, when MMP was at a low state, JC-1could not aggregate but existed as a monomer with green fluorescence. At the end of the proper treatment, the cells were incubated with JC-1 in buffer solution at 37 °C for 20 min and washed twice with FBS-free medium. The fluorescence intensities of PC12 cells were analyzed by laser scanning confocal microscope.

### 3.12. Measurement of Intracellular MDA Content and SOD Activity 

The content of intracellular MDA and the SOD activity of PC12 cells were measured according to the manufacturer instructions. Briefly, after treatment, the cells were lysed and collected on ice. Subsequently, the cell lysate was centrifuged at 12,000× *g* at 4 °C for 5 min and the supernatants were collected. Protein concentrations of samples were determined by BCA assay. The absorbances of detecting MDA and SOD were respectively read at 532, 450 nm in a microplate reader. The MDA content was expressed in micromoles per mg protein, and SOD activities were expressed as units per mg of total protein.

### 3.13. Stability of Synthetic Peptides against Heat, pH, and Simulated GI Digestion

Thermal stability of synthetic peptides was determined according to the method of Yarnpakdee et al. [41] with minor modifications. The synthetic peptides were respectively incubated at 25, 60, 70, 80, 90 °C for 30 min and then measured their scavenging ability of ABTS cation radical. The ability of synthetic peptides against pH treatments was measured according to Zhu et al. [42]. In brief, pH of peptide solution was adjusted to 1, 3, 5, 7, 9, 11, incubated at room temperature for 1 h, and then ABTS cation radical scavenging activity of the treated peptides was measured. A two-stage digestion of protease in vitro (pepsin-digestion for 1 h and then pancreatin-digestion for 2 h at 37 °C) was performed as described by Zhang et al. [43]. The ABTS cation radical scavenging ability was evaluated to assess the peptides stability against GI digestion.

### 3.14. Statistical Analysis

All the experiments were performed in triplicate and results were expressed as means ± SD. Statistical evaluation was performed using one-way ANOVA with the GraphPad Prism 5 software (GraphPad Software, San Diego, CA, USA). *P* < 0.05 was considered to be statistically significant.

## 4. Conclusions

In this study, two antioxidant peptides (DR12 and AR12) were purified and identified from housefly pupae hydrolysate. Both peptides exhibited potent antioxidant activity to remove the various radicals and neuroprotective capacity against H_2_O_2_ induced oxidative stress damage in PC12 cells, and AR12 can significantly decrease intracellular ROS and MDA content, recover cellular MMP, increase the activity of intracellular SOD, and further protect PC12 cells against H_2_O_2_-induced apoptosis and oxidative injury. In addition, antioxidant activity of DR12 is stable against thermal treatments but decreases when treated by acid and alkali, or by GI digestion. AR12 exerts much better ability to remove ABTS cation radical when treated by heating or adjusted to pH acidic and alkaline, besides, AR12 shows great stability against GI digestion. Overall, both peptides have a chance to be potential antioxidants or precursors to antioxidants, and AR12 might be applied in the field of neuroprotection. All of the results above provide sufficient evidences for the further exploration of antioxidant activity of housefly pupae. 

## Figures and Tables

**Figure 1 molecules-24-04486-f001:**
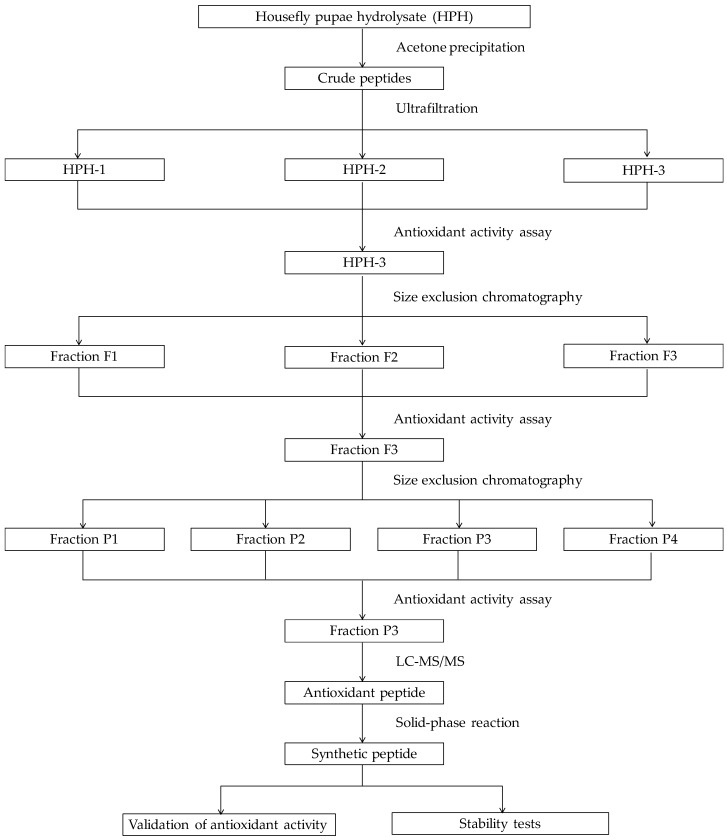
The schematic diagram for this study.

**Figure 2 molecules-24-04486-f002:**
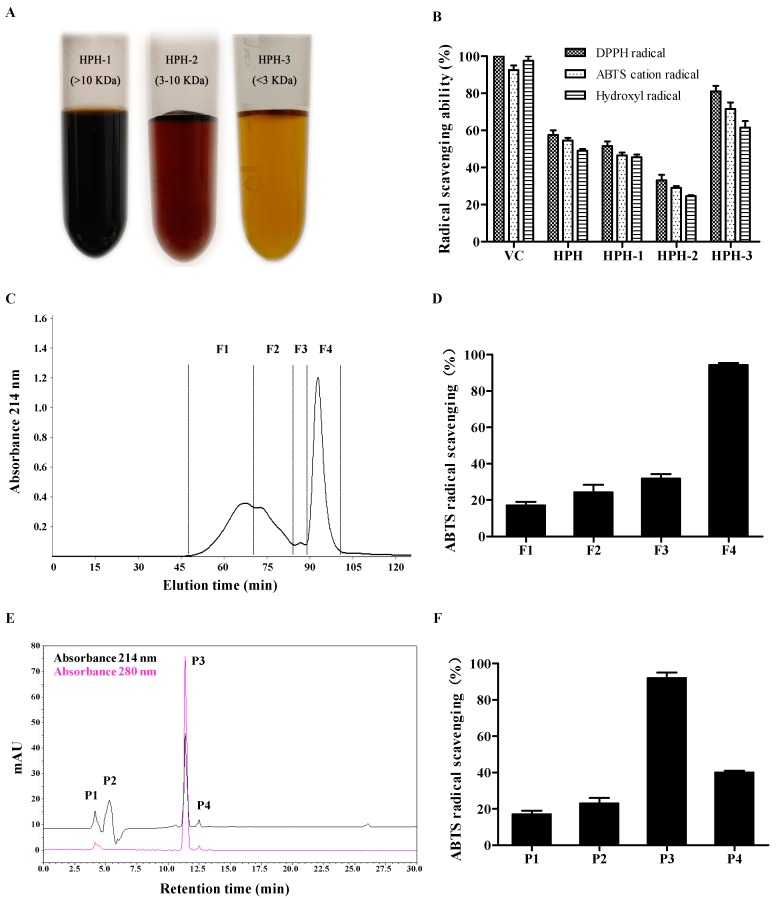
Antioxidant effects of peptide fractions separated from alcalase hydrolysate using UF, SEC and RP-HPLC. (**A**) Three fractions obtained using UF membranes with MWCO’s of 10 kDa and 3 kDa. (**B**) Effects of HPH-1, HPH-2 and HPH-3 ultrafiltration fractions on radical scavenging activity. (**C**) SEC chromatography profile of <3 kDa fraction. (**D**) Effects of F1, F2, F3 and F4 on ABTS cation radical scavenging activity. (**E**) RP-HPLC chromatography profile of F4 fraction. (**F**) Effects of P1, P2, P3 and P4 on ABTS cation radical scavenging activity.

**Figure 3 molecules-24-04486-f003:**
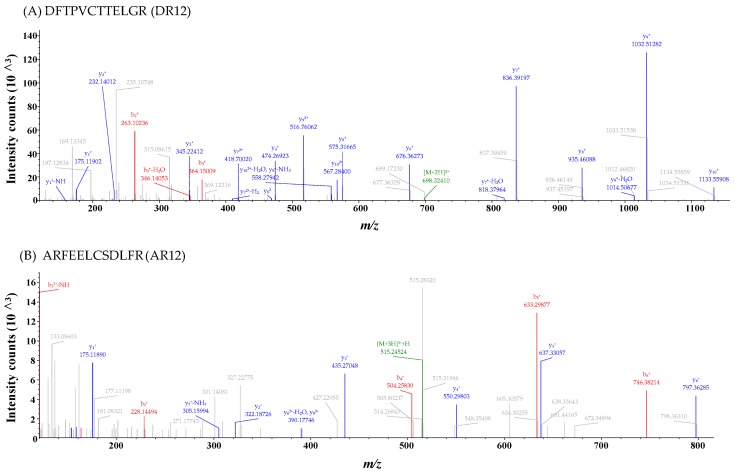
MS/MS spectra of the two peptides (**A**) DR12 and (**B**) AR12.

**Figure 4 molecules-24-04486-f004:**
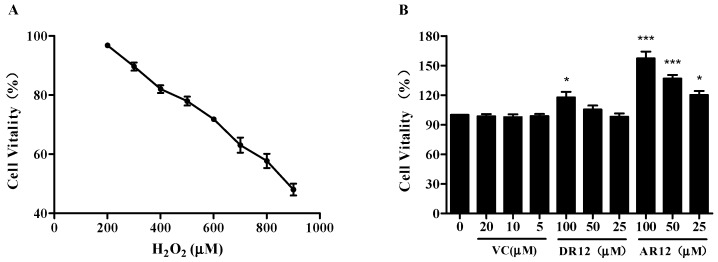
DR12 and AR12 protect PC12 cells against H_2_O_2_-induced cytotoxicity. (**A**) Cells were incubated with different concentrations of H_2_O_2_ (200–900 µM) for 2 h, and then the effect of H_2_O_2_ on cell viability was detected. (**B**) Effect of DR12 and AR12 on PC12 cell viability. Cells were incubated with different concentrations of peptides (25, 50, 100 µM) for 24 h. (**C**,**D**) Effects of DR12 and AR12 (25, 50, 100 µM) on PC12 cells against H_2_O_2_-induced cell death. Cells were pretreated with indicated doses of DR12 for 24 h and then treated with 800 µM H_2_O_2_ for 2 h. VC (10 µM) was used as a positive control. ^###^
*P* < 0.001 compared to control group. * *P* < 0.05, *** *P* < 0.001 compared to model group.

**Figure 5 molecules-24-04486-f005:**
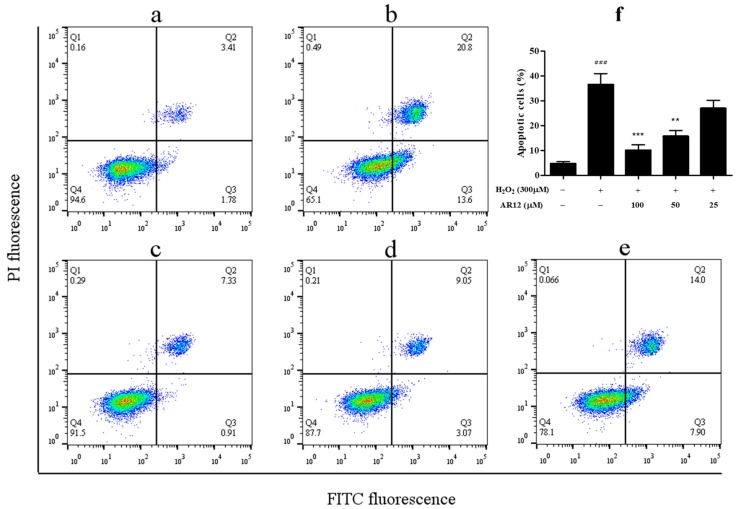
Apoptosis of PC12 cells detected by flow cytometry. (**a**) Apoptosis of normal PC12 cells. (**b**) Apoptosis of the cells treated with 300 µM H_2_O_2_ for 2 h. (**c**–**e**) Apoptosis of the cells that were pretreated with AR12 (100, 50, 25 µM) for 24 h and then treated with 300 µM H_2_O_2_ for 2 h. (**f**) Statistics of apoptosis rate of each group. ^###^
*P* < 0.001 compared to control group. ** *P* < 0.01, *** *P* < 0.001 compared to model group.

**Figure 6 molecules-24-04486-f006:**
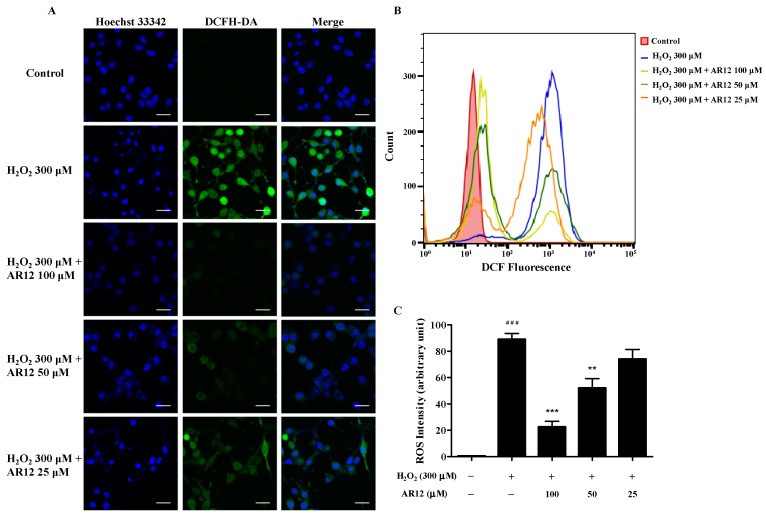
The ROS levels of PC12 cells. (**A**) The levels of ROS were observed by a confocal microscope with DCFH-DA as fluorescent probe. (**B**) The levels of ROS were detected by flow cytometry with DCFH-DA as fluorescent probe. (**C**) Statistics of ROS levels of each group. ^###^
*P* < 0.001 compared to control group. ** *P* < 0.01, *** *P* < 0.001 compared to model group.

**Figure 7 molecules-24-04486-f007:**
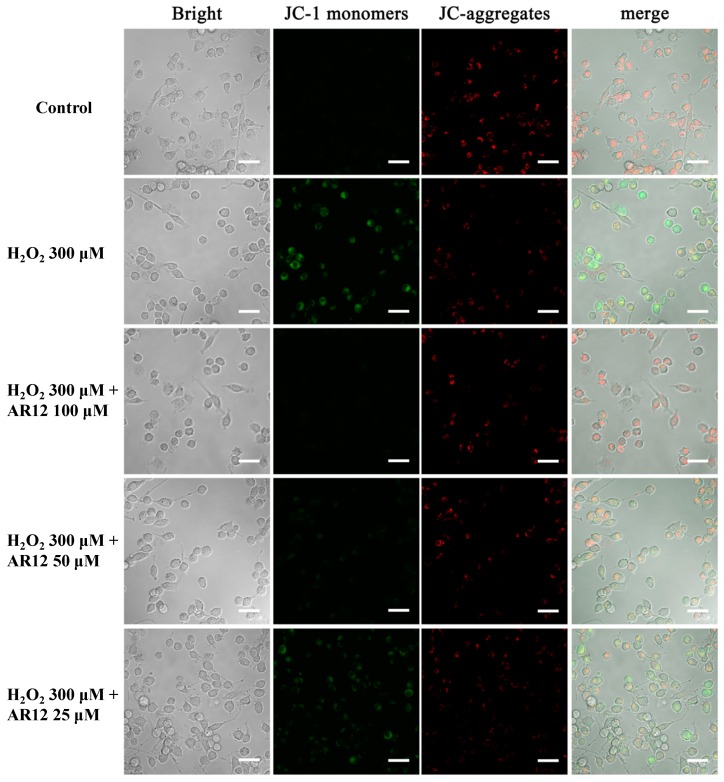
AR12 recovered the loss of MMP of PC12 induced by H_2_O_2_.

**Figure 8 molecules-24-04486-f008:**
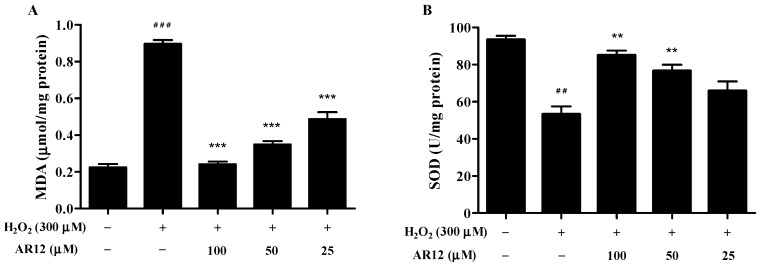
Effects of AR12 on MDA levels and SOD activities in H_2_O_2_-induced PC12 cells. Cells were pretreated with AR12 for 24 h and then treated with 300 µM H_2_O_2_ for 2 h. (**A**) The MDA levels in H_2_O_2_-induced PC12 cells. (**B**) The SOD activities of H_2_O_2_-induced PC12 cells. ^##^
*P* < 0.01, ^###^
*P* < 0.001 compared to control group. ** *P* < 0.01, *** *P* < 0.001 compare to model group.

**Figure 9 molecules-24-04486-f009:**
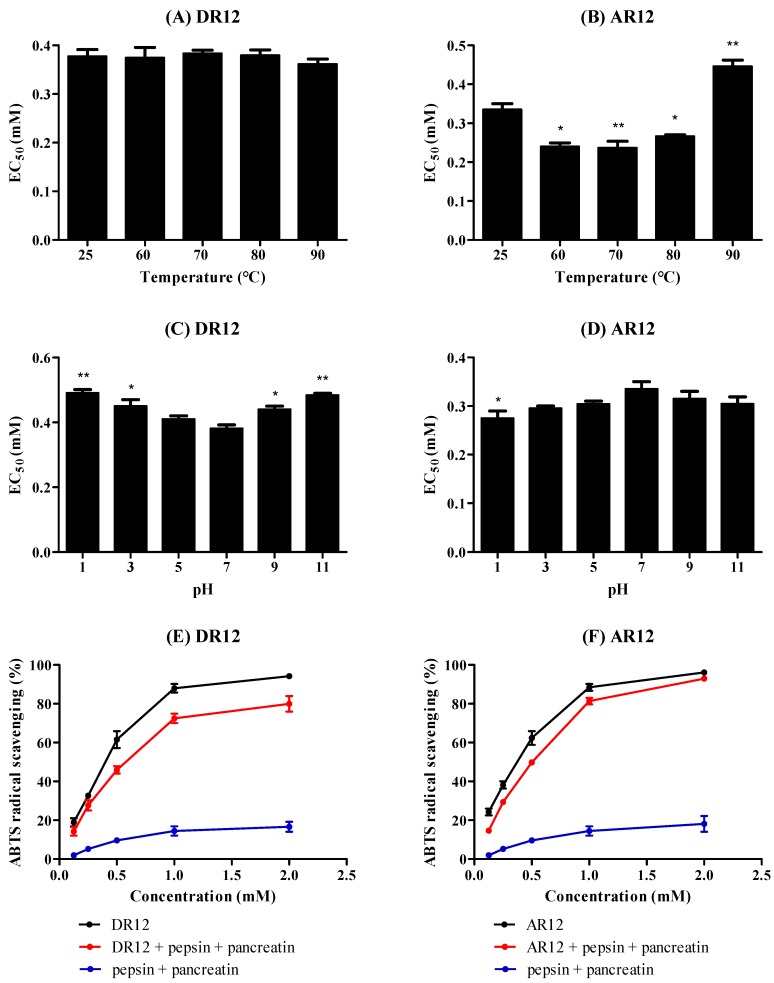
Stability tests of DR12 and AR12. (**A**,**B**) Effects of heating on DR12 and AR12. (**C**,**D**) Effects of pH on DR12 and AR12. (**E**,**F**) Effects of simulated GI digestion on DR12 and AR12. * *P* < 0.05, ** *P* < 0.01 compared to the group of 25 °C or the group of pH 7.

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
