# Peer review of "Housefly Pupae-Derived Antioxidant Peptides Exerting Neuroprotective Effects on Hydrogen Peroxide-Induced Oxidative Damage in PC12 Cells"

_molecules, 2019, doi:10.3390/molecules24244486_

Round 1

Reviewer 1 Report

The authors investigated two antioxidant peptides from the alcalase-hydrolysate of housefly (Musca domestica L.) pupae (DR12 and AR12). Housefly larvae and pupae, namely wuguchong in traditional Chinese medicine, are rich in biomass with high-quality proteins. Both peptides exhibited potent antioxidant activity to remove the various radicals and neuroprotective capacity against H2O2-induced oxidative stress damage in PC12 cells,  and AR12 might should be indicated to be applied in neuroprotection.
Minor suggestions:   1) Introduction. At line 42: which means "reproductive toxicity"? 2) Results and Discussion. At pg. 5, line 143: when the authors states that “This is the first time to identify antioxidant peptides from housefly pupae”. Seeing some papers that studied in similar manner, the potential identification and antioxidant properties of housefly larvae, I recommend you to discuss the particularities of this study using the pupae, showing advantages in comparison using larvae (see below some of these papers):   Wang J, Wang Y, Dang X, Zheng X, Zhang W. Housefly larvae hydrolysate: orthogonal optimization of hydrolysis, antioxidant activity, amino acid composition and functional properties. BMC Res Notes. 2013;6:197. Published 2013 May 17. doi:10.1186/1756-0500-6-19.   Huan ZHang, Pan Wang, ai-Jun ZHang, Xuan Li, Ji-Hong ZHang, Qi-Lian Qin and Yi-Jun Wu. ANTIOXIDANT ACTIVITIES OF PROTEIN HYDROLYSATES OBTAINED FROM THE HOUSEFLY LARVAE. Acta Biologica Hungarica 67(3), pp. 236–246 (2016) DOI: 10.1556/018.67.2016.3.2   3) Results: Pg. 10, H2O2 300 μM + AR12 100 μM: I had the impression of visualizing some cells that, even with positivity for Mitochondrial Membrane Potential activity, are looking like apoptotic cells. Please, discuss.   4) Results: Pg. 12, Fig. 9E and F: the three symbols used in these graphics are so small and similar, that difficult us to compare.   5) Material and Methods: Pg. 14, “3.8. Cell culture and viability analysis”: the H2O2 (800 μM) concentration informed is incorrect, once the dose-response curve was plotted with different concentrations of H2O2 (200-900 μM) for 2 h, and not 1 h.   6) Material and Methods: Pg. 15, “3.9. Apoptosis assay”: the legend of Fig. 5 informs that the H2O2 treatment (300 μM) was performed for 2 h.   Please, check all details on Materials and Methods, comparing with the information on the legends of each figure.  

Author Response

Dear Reviewer:

    Thank you for your comments concerning our manuscript entitled “Housefly Pupae-Derived Antioxidant Peptides Exerting Neuroprotective Effects on Hydrogen Peroxide-Induced Oxidative Damage in PC12 Cells” (ID: molecules-653428). Those comments are all valuable and very helpful for revising and improving our paper, as well as the important guiding significance to our researches. We have studied comments carefully and have made correction which we hope meet with approval. Revised portions are marked in red in the paper (please see the attachment). Responses to the comments in blue as following:

Introduction. At line 42: which means "reproductive toxicity"?

Reproductive toxicity means the toxic effects on reproductive function and development process of mammals. In detail, it means the toxic effects towards germ cells, conception, pregnancy, delivery and lactation in parental generation, and toxic effects to development of embryo, fetus or infant. We are very sorry for the loose presentation, according to the reference 12, butylated hydroxyanisole and butylated hydroxytoluene have developmental but not reproductive toxicity to the early life stage of zebrafish. However, previous studies indeed reported that some synthetic antioxidants have reproductive toxicity to mice or rats (Ham, J.; Lim, W.; Park, S.; Bae, H.; You, S.; Song, G. Synthetic phenolic antioxidant propyl gallate induces male infertility through disruption of calcium homeostasis and mitochondrial function. Environ. Pollut. 2019, 248, 845–856. DOI: 10.1016/j.envpol.2019.02.087). Therefore, we add “developmental toxicity” at the site of reference 12 and add a new reference at the site of “reproductive toxicity” to perfect our expression. At the same time, we have checked the other references again and made some changes which have been marked red.

I recommend you to discuss the particularities of this study using the pupae, showing advantages in comparison using larvae.

Our previous study displayed that the species and contents of protein are obviously different among various lifespan stage of housefly, and protein accumulates from egg to adult, namely that protein in pupae is much richer than that in larvae. Therefore, the activity and active substances may be different between larvae and pupae. However, the exploration of activity or active components of housefly pupae is almost blank to date. So, we carried out this study to support the development of housefly pupae. Compared to the previous studies, we are the first to identify antioxidant peptides in housefly pupae. In the introduction of manuscript, we have also enhanced the differences of larvae and pupae which have been marked red.

Results: Pg. 10, H2O2 300 μM + AR12 100 μM: I had the impression of visualizing some cells that, even with positivity for Mitochondrial Membrane Potential activity, are looking like apoptotic cells. Please, discuss.

We are sorry for the flawed photos that a few cells in control group and the group treated with 100 μM AR12 are apoptotic. We ought to be responsible for our negligence in taking the two photos. However, there are some reasons for this phenomenon. Firstly, too many preprocessing steps of measuring these cell indexes might be the primary cause leading to apoptosis, and then apoptosis is a natural physiological process of cells as well. These are also the reasons that there are about 5% apoptotic cells in control group in the apoptosis assay.

Results: Pg. 12, Fig. 9E and F: the three symbols used in these graphics are so small and similar, that difficult us to compare.

Considering the Reviewer’s suggestion, we have changed the color of curve to make it clear.

Material and Methods: Pg. 14, “3.8. Cell culture and viability analysis”: the H2O2 (800 μM) concentration informed is incorrect, once the dose-response curve was plotted with different concentrations of H2O2 (200-900 μM) for 2 h, and not 1 h.

We are very sorry for our negligence, we have revised the concentration and incubation time (2 h) of H2O2. Furthermore, we also checked the full text again to find out and correct other simiar mistakes.

Material and Methods: Pg. 15, “3.9. Apoptosis assay”: the legend of Fig. 5 informs that the H2O2 treatment (300 μM) was performed for 2 h. Please, check all details on Materials and Methods, comparing with the information on the legends of each figure.

Thanks for your careful help in finding out the problems in the manuscript, we have checked the full text and revised the inaccurate details which have been marked red in the manuscript.

    We tried our best to improve the manuscript and made some changes in the manuscript. These changes will not influence the content and framework of the paper. And here we did not list the changes but marked in red in revised paper.

    We appreciate for your work earnestly, and hope that the correction will meet with approval.

Sincerely yours,

Dr. Depo Yang

Professor

School of Pharmaceutical Sciences, Sun Yat-sen University, Guangzhou, China,

510006.

Telephone: +86 020-39943043

Reviewer 2 Report

The article deals with the characterization of two peptides with antioxidant activity. Data and results are consistent and well presented.

The work will be improved if the conclusions are strengthened and if in the introduction section the reasons why the work was undertaken are enhanced.

Author Response

Dear Reviewer:

    Thank you for your comments concerning our manuscript entitled “Housefly Pupae-Derived Antioxidant Peptides Exerting Neuroprotective Effects on Hydrogen Peroxide-Induced Oxidative Damage in PC12 Cells” (ID: molecules-653428). These comments are very helpful for revising and improving our paper, as well as the important guiding significance to our researches. We have studied comments carefully and have made correction which we hope meet with approval. Revised portions are marked in red in the paper (please see the attachment). Responses to the comments in blue as following:

The work will be improved if the conclusions are strengthened and if in the introduction section the reasons why the work was undertaken are enhanced.

To meet the reviewer’s comments, the conclusion and introduction section were carefully checked and improved in the revised manuscript.

    We appreciate for your work earnestly, and hope that the correction will meet with approval.

Sincerely yours,

Dr. Depo Yang

Professor

School of Pharmaceutical Sciences, Sun Yat-sen University, Guangzhou, China,

510006.

Telephone: +86 020-39943043
